## PROCEEDINGS A

computer science, quantum computing, quantum physics

simulated universes, existential risk, eavesdropping

**Authors for correspondence:**
Alexandre Bibeau-Delisle
e-mail: alexbibdel@gmail.com
Gilles Brassard
e-mail: brassard@iro.umontreal.ca

# Probability and consequences of living inside a computer simulation

Alexandre Bibeau-Delisle[1] and Gilles Brassard FRS[1,2]

[1]Département d'informatique et de recherche opérationnelle, Université de Montréal, C.P. 6128, Succursale Centre-Ville, Montréal, QC H3C 3J7, Canada
[2]Canadian Institute for Advanced Research, Toronto, ON M5G 1M1, Canada

AB-D, 0000-0001-5343-7044; GB, 0000-0002-4380-117X

It is shown that under reasonable assumptions a Drake-style equation can be obtained for the probability that our universe is the result of a deliberate simulation. Evaluating loose bounds for certain terms in the equation shows that the probability is unlikely to be as high as previously reported in the literature, especially in a scenario where the simulations are recursive. Furthermore, we investigate the possibility of eavesdropping from the outside of such a simulation and introduce a general attack that can circumvent attempts at using quantum cryptography inside the simulation, even if the quantum properties of the simulation are genuine.

## 1. Introduction

The question of whether or not we are living inside a computer simulation has inspired a large amount of fiction (notably the novel *Simulacron-3* [1] and the movie *The Matrix* [2]), but, unsurprisingly, not much serious research. Among the more reasonable and quantitative attempts, let us mention Nick Bostrom's simulation argument [3]: if societies do not tend to self-destruct before acquiring the technology necessary to the exploitation of a significant fraction of the computing power inherently permitted by the laws of physics, our probability of living inside a simulation approaches unity. This fairly pessimistic point of view has been widely publicized, for instance making it into *The Guardian* [4], where, among others, Elon Musk is reported

to have ascertained that we almost certainly live within a simulation, going as far as saying 'The odds we're in base reality is one in billions' at the Code Conference 2016 [5]. Despite some of our arguments being compatible with such a view, we show that the cost of simulating our environment and the possibility of recursive simulations indicate that in most realistic scenarios the odds are instead in favour of our living in base reality.

As with many things in computer science, the idea that our world may be a simulation needs revisiting in light of the development of quantum computing. While it remains to be formally proven that quantum computers are more powerful than their classical counterparts (the so-called BQP ≠ BPP conjecture), accumulating evidence seems to support the intuition that many quantum phenomena cannot be efficiently simulated with classical computing power, often incurring an exponential slowdown, as famously noted by Richard Feynman as early as 1981 [6]. We shall thus proceed on the assumption that the current scientific consensus is correct and that simulating the whole of our physics on classical resources would be infeasible. At the other end of the spectrum, we shall suppose that our physics can be efficiently simulated on quantum computing resources that we can theoretically envision. This is in fact not necessarily true, for instance if there is no scale at which physics becomes discrete and the information density of our world is infinite or if gravity is truly beyond quantum mechanics and requires an even more powerful computation paradigm. After using these assumptions to estimate the proportion of beings potentially resulting from deliberate simulations, we investigate the possibility of eavesdropping from the outside of such a simulation. While it is not possible to perform arbitrary measurements on a genuinely quantum simulation without disturbing it, we show that the combination of the essentially classical nature of our thoughts and the possibility of the simulation being adaptively rewired gives the simulators an insurmountable advantage.

## 2. Estimating the proportion of simulated beings

The computing power theoretically attainable with the known laws of physics is immense. Harnessing this power from a single kilogram of matter would allow for roughly $10^{50}$ OPS (operations per second) [7], and furthermore these could be quantum operations on qubits. Let us denote this computing power density $D_{Mat} = 10^{50}$ OPS/kg. By comparison, reasonable estimates on $P_{Bra}$, the computing power of the human brain (whose average mass is $M_{Bra} \approx 1.4$ kg [8]), vary between $10^{14}$ and $10^{16}$ OPS [9,10]. Here we use $P_{Bra} = 10^{16}$ OPS. In a hypothetical civilization with a technological level (which we denote Civ) sufficient to use a significant proportion of the computing power intrinsic to physical matter, say one-billionth, it may thus be possible for a single computer the mass of a human brain to simulate the real-time evolution of $1.4 \times 10^{25}$ virtual brains. We recognize that anthropocentric comparisons with the human brain are totally arbitrary on the cosmic scale, but exact numbers are inconsequential for our purpose. Furthermore, as we are investigating the whole question from a philosophical and futuristic point of view, we need not worry about the fact that our own civilization is extremely far from being able to develop the technology we are discussing.

In a Civ-level civilization, each individual might reasonably be able to exploit such awesome computing power. Of course, the actual amount of available computing power will vary immensely from civilization to civilization and from individual to individual. For our treatment, it is sufficient that this amount be bounded, as a result of limitations stemming from either the dynamics of advanced civilizations or from the laws of physics themselves.[1] Indeed, while an advanced individual might conceivably monopolize a whole planet and transform it into a giant supercomputer, they could certainly not do the same with a whole cluster of galaxies, if only because of the no-signalling principle. We denote $M_{Use}$ the average equivalent mass in maximally harnessed matter that each individual in Civ can use for their computations, meaning that their

[1]On a universal scale, this will result in a certain proportionality between the total population and the amount of computing power monopolized. All other parameters being the same, a universe twice as large as another will on average have both twice its total population and twice its computing power.

computational power is given by $M_{Use}D_{Mat}$. To cast this in units of the number of brains they could simulate in real-time, we use the computational power ratio

$$R_{Cal} = \frac{M_{Use}D_{Mat}}{P_{Bra}}. \tag{2.1}$$

(In order to help keep track of the many symbols used in this paper, the symbols are listed in table 1 in appendix A.)

Over the history of a civilization and its ancestors, we denote $f_{Civ}$ the fraction of individuals having access to Civ-level computing power (individuals living before the development of such technology, like ourselves, fall under $1 - f_{Civ}$). We also denote $f_{Ded}$ the proportion of the computing power available to people in $f_{Civ}$ dedicated to simulating virtual consciousnesses. We may now use these (unknown) factors to obtain a very rough first estimate of $N_{Sim}$, the number of sentient beings living in a simulated world

$$N_{Sim} = N_{Re}f_{Civ}f_{Ded}R_{Cal}, \tag{2.2}$$

where $N_{Re}$ is the number of individuals living in the real world. We then obtain the fraction of the total sentient population that is real as

$$f_{Re} = \frac{N_{Re}}{N_{Re} + N_{Sim}} = \frac{1}{1 + f_{Civ}f_{Ded}R_{Cal}} \tag{2.3}$$

and the fraction that is simulated as

$$f_{Sim} = 1 - f_{Re} = \frac{f_{Civ}f_{Ded}R_{Cal}}{1 + f_{Civ}f_{Ded}R_{Cal}}. \tag{2.4}$$

This may be generalized to the complete set of civilizations (each denoted by the index $j$, having total population $N_{Totj}$, counting both real and simulated beings, and its own value for each factor in equation (2.3)) over the life of the universe. The universal proportion of real consciousnesses is then

$$f_{ReU} = \frac{\sum_j f_{Rej}N_{Totj}}{\sum_j N_{Totj}}. \tag{2.5}$$

While the mathematical formulation in equation (2.4) is more general than Bostrom's [3], the basic idea, which we challenge here, remains the same. Recast in our scenario, the original simulation argument says that, given that $R_{Cal}$ is gigantic (even once we factor in limited efficiency—see below—or lower the average amount of matter $M_{Use}$ that is effectively leveraged per individual in Civ-level civilizations), one of the following statements must be true:

(i) The probability that we live in a simulation approaches unity, i.e. $f_{Sim} \approx 1$, or
(ii) $f_{Civ}f_{Ded} \approx 0$, i.e. at least one of $f_{Civ}$ or $f_{Ded}$ is vanishingly small.

It is mathematically inescapable from equation (2.4) and the colossal scale of $R_{Cal}$ that $f_{Sim} \approx 1$ unless $f_{Civ}f_{Ded} \approx 0$ (since $x/(1+x)$ approaches 1 when $x$ becomes very large). Nevertheless, one could reasonably challenge the claim made surreptitiously in statement (i) above. Indeed, is the probability that we live in a simulation accurately represented by $f_{Sim}$? This claim assumes implicitly that our world is typical, taking no account of possible extra evidence to the contrary. In particular, it assumes that simulated intelligent beings are just as conscious as real ones, which is certainly logical in the case where the necessary biological processes are perfectly simulated, letting intelligence arise 'naturally', rather than having preprogrammed artificial intelligence, no matter how complex. More importantly, it supposes that the quality and persistence of our world and the lack of inconsistencies in its behaviour can be disregarded as evidence that it is real. In fact, $f_{Sim}$ is only a baseline probability that needs to be adjusted according to Bayes' inference rules in the light of other factors, which are virtually impossible to evaluate precisely. Nevertheless, we concede that $R_{Cal}$ is sufficiently large for this caveat not to invalidate Bostrom's simulation argument [3]. Instead, the main thrust of the rest of this section is to challenge equation (2.4) itself, and therefore the ominous inevitability of $f_{Civ}f_{Ded} \approx 0$ whenever $f_{Sim}$ is not close to unity.

But first, let us pretend we accept equation (2.4) and explore the possibility that $f_{Civ}f_{Ded}$ indeed approaches 0. If $f_{Ded} \approx 0$, it simply means that advanced civilizations do not use a significant portion of their computing power to simulate worlds like ours. This could happen for a large variety of reasons, from a simple lack of interest to a social taboo. A society of beings similar to us (but with a much greater technological development) could indeed decide it is not very ethical to simulate beings with enough precision to make them conscious while fooling them and keeping them cut-off from the real world. On the other hand, we can imagine many motivations that would lead them to create such simulated worlds and beings: sociological research, strategic planning, and maybe even an advanced form of exotic tourism. Indeed, someone with enough resources to simulate a world and its beings could also possess the technology necessary to project themselves into the simulation in order to explore different planets, eras, and realities. Even if laws or taboos seek to prevent the simulation of complete consciousnesses, infrequent infractions would be sufficient to increase $f_{Ded}$, and therefore $f_{Sim}$ (unless $f_{Civ} \approx 0$, which we explore in the next paragraph), to a non-negligible figure.

If instead $f_{Civ} \approx 0$, the conclusion might be considerably more worrisome. It would suggest that societies of intelligent beings are unable to reach the Civ level. It is indeed possible that it is technologically much easier to create weapons capable of eradicating entire civilizations in the real world than to obtain the computing power needed to simulate them in a virtual world. This is probably what Musk meant when he said at the Code Conference 2016: 'Either we're going to create simulations that are indistinguishable from reality, or civilization will cease to exist. Those are the two options.' [5] Even if civilizations become more mature and peaceful as they evolve, as scientific advances multiply not only the computational but also the destructive capabilities of individuals, isolated instances of murderous insanity might be sufficient to lead societies to extinction. The dramatic increase in both types of power the human species has achieved since World War II has probably brought us closer to self-eradication than to world simulation, but how it plays out in the future remains to be seen. Furthermore, we can only speculate about other civilizations and species.

Does equation (2.4) condemn us to a pessimistic vision of the world where we are either prisoner of someone else's simulation (and thus at their mercy), or on a path towards assured self-destruction? As the equation is considerably too simple to account for all important factors, not necessarily. For instance, it is not sufficient to be able to simulate a large amount of virtual brains in order to obtain a credible simulation of a world resembling ours: the environment must also be adequately simulated. Bostrom [3] discounts the computational cost of simulating the environment based on the fact that the human sensory bandwidth is on the order of $10^8$ bits per second and that physics can be brought down to a minimum degree of complexity in a simulation. However, $10^8$ bits per second per person is merely the size of the input to the human senses and does not account for the complexity of the computation required to obtain the correct bits that will consistently fool all the simulated consciousnesses (not only into thinking they are 'real' but also into thinking the known laws of physics are properly respected. Saying that this is computationally easy because the human senses have limited bandwidth is akin to saying solving chess is easy because there are only three possible outputs (white wins, black wins, or draw).

To account for and investigate the cost of simulating the environment, we introduce the factor $C_{Env}$ into equation (2.3), and thus implicitly also into equation (2.4). We define $C_{Env}$ as the average ratio of the computing power necessary to simulate the environment of an individual to that necessary to simulate his consciousness. As no computation is 100% efficient, we also introduce $f_{Eff}$ to represent the reciprocal of the number of physical operations required in base reality to perform one logical operation in the simulation. Thus, the number of simulated individuals $N_{Sim}$ and the proportion of real beings $f_{Re}$, previously given by equations (2.2) and (2.3), must be amended as follows:

$$N_{Sim} = \frac{N_{Re}f_{Civ}f_{Ded}R_{Cal}f_{Eff}}{1 + C_{Env}} \qquad (2.6)$$

and

$$f_{\text{Re}} = \frac{N_{\text{Re}}}{N_{\text{Re}} + N_{\text{Sim}}} = \frac{1}{1 + \frac{f_{\text{Civ}} f_{\text{Ded}} R_{\text{Cal}} f_{\text{Eff}}}{1 + C_{\text{Env}}}}. \tag{2.7}$$

We see that $f_{\text{Re}}$ increases compared to equation (2.3). Let us look at different scenarios to see just how much, depending on the environment's simulated physics.

Physics could of course be completely different inside and outside the simulation. The level of complexity needed in the laws of physics of the simulation depends very much on its purpose. If the interest is simply to create worlds where many intelligent beings live and interact, it would be advantageous to choose laws of physics that allow for intelligence but minimize the cost of simulating the environment. This would be the case for simulated worlds designed for tourism, gaming, and general escapism. However, by their very nature, these are unlikely to require quantum physics and, as we are investigating these scenarios to evaluate the probability that our own world is a simulation, we note that if our physics had been chosen to minimize the required computing power while providing an entertaining environment, we almost certainly would not find such a high ratio of $M_{\text{Bra}} D_{\text{Mat}}$ to $P_{\text{Bra}}$. On the other hand, for more 'serious' purposes, civilizations are likely to be more interested in simulating societies that resemble them and evolve in similar environments, as lessons learned from the simulations would be much easier to apply to the real world. Indeed, when a civilization has the Civ level of technology, a large part of its dynamics must be strongly dependent on science and technology and a simulation of a world with different laws of physics and technological possibilities is likely to quickly diverge from it. Thus, while the quantum nature of our universe, and probably of the computing power necessary to simulate it, does not change our equations (which would be as valid for classical worlds simulated on classical computers, provided their classical laws of physics can still support a gigantic $R_{\text{Cal}}$), it restricts the type of simulation in which we could likely be. This comes into play as we consider what happens when simulated worlds behave similarly to those from which they are simulated.

If instead of being chosen for low computational complexity, the laws of physics are similar inside and outside the simulation, it would be extremely costly to have the environment simulated down to its most microscopic level. Indeed, interactions leading to consciousness inside our brains represent only a minuscule fraction of all individual interactions taking place in the environment, leading to a $C_{\text{Env}}$ so large that even the scale of $R_{\text{Cal}}$ cannot compensate for it and allow the simulation of a large number of intelligent beings.

In order to be both tractable and representative of the real world, a simulation should have a variable level of complexity, cutting corners while no intelligent being is paying close attention, but able to recreate the full complexity of physics when necessary. In this type of scenario, we would be forcing our simulators to temporarily use more computing power to simulate our environment when we conduct experiments that enable us to understand (or exploit) physics at its microscopic level. On the flip side, when we are being inattentive to physics on that scale, the computing power required to simulate our environment could indeed be much smaller. A simulated universe with a variable level of complexity could provide an explanation to Fermi's paradox.[2] Indeed, while advanced intelligent civilizations could build self-replicating probes that enable a relatively fast search of a galaxy (over perhaps a few million years even if superluminal travel is impossible), we shall never encounter such probes if we live in a simulation in which the default physics is too simplified for life to arise far away from the (simulated) Earth. Actually, the fact that we have not detected any evidence for the existence of extraterrestrial civilizations may be considered as the most convincing argument *in favour* of the theory according to which we live in a simulation, quite unlike other arguments introduced in this paper, which run in the opposite direction. Note that while a similar argument could be made for regions of the Earth that interact little with humans (deep sea ecosystems, for instance), they remain coupled with us much more

---

[2]The fact that we detect no evidence for the existence of extraterrestrial civilizations despite the great number of star systems in which they could potentially arise [11,12].

strongly than things outside our solar system and their simulated complexity could not be cut down nearly as much without significantly affecting the progression of the simulation.

Now that we have argued that a simulation with variable complexity is most advantageous (and therefore most likely) from the simulators' point of view, we turn to the possibility of recursive scenarios.[3] Considering that the simulated world should be similar to the real one to maximize its utility, civilizations within it are likely to start creating their own simulations once they have reached Civ. If the initial simulation features physics with a complexity that varies depending on how the simulated beings use or observe it, it will slow down considerably when they start using a large amount of computing power in order to run their own simulations. This should remain undetectable from inside the simulation, as it is only possible to compare the relative speeds of phenomena within one's self or environment, and a uniform slowdown compared to an external reference has no impact on the simulated denizens. However, from the point of view of the (real) simulators, their simulation will gradually slow down unless it is allotted more and more computing power. This will happen even more dramatically if the simulated civilization uses a large amount of computing power for various purposes, in addition to running their own simulations. As unavoidable consequence of the simulated beings' computations, a significant increase in $C_{Env}$ will take place in equations (2.6) and (2.7).

To estimate this increase, recall that a proportion $f_{Civ}$ of the simulated population will have reached a technological level sufficient to harness a computing power equivalent to $M_{Use}D_{Mat}$ per individual. If all this available power is actually used, each of these simulated individuals will incur an environmental simulation cost equivalent to $M_{Use}D_{Mat}/P_{Bra} = R_{Cal}$ (by equation (2.1)) times the cost of the simulation of their own existence (in addition to the computational cost of the rest of their environment). This environmental overhead ratio is precisely what we called $C_{Env}$. It follows that

$$C_{Env} \geq f_{Civ}R_{Cal} \tag{2.8}$$

in a scenario according to which the simulated beings use all their available computing power once they have reached technological level Civ. But even if not all the available computing power is used by these denizens, at least an $f_{Ded}$ fraction of it *must* be spent by definition for the simulated civilizations' own simulations. We may therefore prefer to use the more conservative (but inescapable) bound

$$C_{Env} \geq f_{Civ}f_{Ded}R_{Cal}. \tag{2.9}$$

If the first simulated level can create its own simulations, the same goes for the second level, the third, and so on. This gives us a scenario in which the simulations are nested within one another (figure 1) and where, of course, the computing power available on a given level must be strictly less than that in the one above it (in the sense of closer to reality). To obtain a mathematical model of this scenario, let $N_i$ denote the 'real' population of level $i$ (unlike $N_{Tot j}$ in equation (2.5)), so that $N_0$ denotes the real (no quotes) population in base reality, which we called $N_{Re}$ previously. We adapt equation (2.6) to the recursive context by replacing $N_{Re}$ and $N_{Sim}$ by $N_i$ and $N_{i+1}$, respectively, which yields the recurrence equation

$$N_i f_{Civ}f_{Ded}R_{Cal}f_{Eff} = N_{i+1}(1 + C_{Env}). \tag{2.10}$$

The parameters in this equation could of course vary between the levels, but we can use average parameters to investigate what happens when all levels are similar. Equation (2.10) enables us to find the average population ratio between consecutive levels, which we call $f_{Pop}$ for simplicity:

$$f_{Pop} = \frac{N_{i+1}}{N_i} = \frac{f_{Civ}f_{Ded}R_{Cal}f_{Eff}}{1 + C_{Env}}. \tag{2.11}$$

It follows immediately that the estimated population $N_i$ at level $i > 0$ is

$$N_i = (f_{Pop})^i N_0, \tag{2.12}$$

---

[3]The idea of a simulation inside a simulation was already at the heart of the story in the 1964 novel *Simulacron-3* [1]!

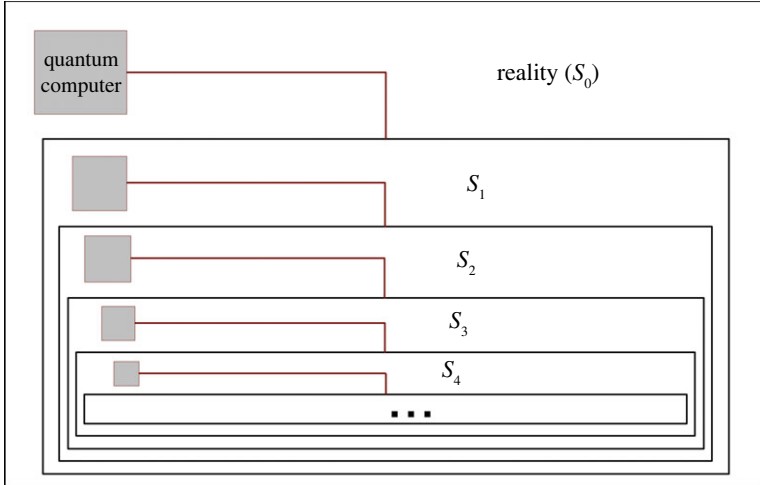

**Figure 1.** Hierarchy of simulations nested within one another. The system on which the level $i + 1$ simulation is computed is on level $i$. The total computing power used to simulate a level must therefore gradually decrease as we get further and further from base reality (level 0). Note that while we show only one simulation per level, in all likelihood there would be many running in parallel. (Online version in colour)

which leads to a geometric series for the total population of a system of maximum simulation depth $I$

$$N_{\text{Tot}} = \sum_{i=0}^{I} N_i = N_0 \sum_{i=0}^{I} (f_{\text{Pop}})^i. \tag{2.13}$$

As the number of levels is large but finite,[4] the infinite geometric series gives us a bound on the sum, which allows us to obtain a new version of equation (2.7)

$$f_{\text{Re}} = \frac{N_0}{N_{\text{Tot}}} = \frac{1}{\sum_{i=0}^{I} (f_{\text{Pop}})^i} \geq \frac{1}{\sum_{i=0}^{\infty} (f_{\text{Pop}})^i} = 1 - f_{\text{Pop}}. \tag{2.14}$$

The total fraction of simulated beings is therefore

$$f_{\text{Sim}} = 1 - f_{\text{Re}} \leq f_{\text{Pop}}. \tag{2.15}$$

Unfortunately, it is extremely difficult to estimate the values of the factors appearing in equation (2.11), which defines $f_{\text{Pop}}$. In a way, the situation is analogous to that of Drake's equation [13], which contains too many hard-to-estimate factors to really enable us to calculate the number of extraterrestrial civilizations in the galaxy. We can nonetheless conclude from equations (2.9) and (2.11) that

$$f_{\text{Pop}} < \frac{f_{\text{Civ}} f_{\text{Ded}} R_{\text{Cal}} f_{\text{Eff}}}{C_{\text{Env}}} \leq \frac{f_{\text{Civ}} f_{\text{Ded}} R_{\text{Cal}} f_{\text{Eff}}}{f_{\text{Civ}} f_{\text{Ded}} R_{\text{Cal}}} = f_{\text{Eff}} \tag{2.16}$$

and it follows from equation (2.15) that $f_{\text{Sim}}$ also is upper-bounded by $f_{\text{Eff}}$, which is most likely under 50%. Indeed, even for an advanced civilization, a quantum computation that requires

---

[4]Note that even in an infinite universe with infinite real and simulated populations, the number of levels must remain finite. Indeed, each level must be a collection of finite simulations, which would be infinite in number in an infinite universe, each involving a finite quantity of energy and computing power. The strictly decreasing available computing power at each further level means that at some point the available power for each individual simulation falls below the minimum required amount to support a complex civilization. While arbitrarily large but still finite 'local' simulations could potentially support arbitrarily deep recursions, if they came to dominate they would simply cause the bound of the infinite geometric series to be attained in equation (2.14), preserving the validity of our equations. Furthermore, note that in order to regularize our treatment in the case of a universe infinite in space, time, or even number of branches, the factors must be evaluated and averaged out over a finite region that is sufficiently large in all dimensions and parameters to be representative.

no more than two physical operations per logical one would be quite an achievement! If we accept equation (2.8), which holds assuming that the simulated beings use all their available computing power once they have reached technological level Civ, the same reasoning allows us to lower our upper bound on $f_{Sim}$ to $f_{Ded}f_{Eff}$, for which there is little possible doubt it would fall below 50%. The same conclusion holds (with no need to assume anything about $f_{Eff}$) even if we do not accept equation (2.8), under the hard-to-contradict assumption that the simulated civilization will not spend more computing power on simulations than on the sum total of all their other computations. Using equation (2.8) and more plausible values, for instance $f_{Eff} = 0.05$ and $f_{Ded} = 0.2$, yields

$$f_{Sim} \leq f_{Pop} < f_{Ded}f_{Eff} = 0.2 \times 0.05 = 1\% \tag{2.17}$$

The recursive scenario thus gives us a reasonable situation in which there are more real than simulated consciousnesses despite the gigantic value of $R_{Cal}$. It is interesting to note that we are able to reach this conclusion by assuming *recursively* that simulated civilizations will start creating their own simulations once they have reached Civ, whereas we would not be able to do so on the basis of equation (2.6) alone. This is somewhat counterintuitive since one might be tempted to think that the recursive creation of simulations can only increase the number of simulated beings!

Equation (2.12) predicts an ever-decreasing probability of existing at a specific level $i$ of the simulation hierarchy as $i$ increases. This is a fairly direct consequence of the fact that the total computing power must decrease with each successive level and that the complexity of the simulated beings or laws of physics cannot keep decreasing. In a sense, this is reassuring, as the deeper one exists in the hierarchy, the higher the probability of the simulation suffering an apocalyptic failure or shutdown becomes. Indeed, if any of the levels experiences a cataclysm, a war, a social reorganization, or simply a loss of interest, it could mean the end of all simulations under it. The long-term persistence of our world (assuming it is not an illusion) is therefore at least evidence that lowers the likelihood that it is a simulation existing very deep in this hierarchy.

Note that it is not necessarily true that at any given time the population of level $i + 1$ is smaller than that of level $i$. The more limited computing power for level $i + 1$ could support a greater amount of consciousnesses if time elapses at a slower rate than on the level above. This still results in a smaller total population for level $i + 1$ once integrated over a sufficiently long reference time external to the simulations. In this scenario, the probability that the simulation suffers catastrophic failure during the lifetime of an individual increases even faster as a function of $i$. Indeed, the compounded effect of the slowdown factors multiplied together makes it so that during a simulated being's lifespan, entire civilizations could rise and fall on levels much closer to reality, greatly lowering the chance of uninterrupted simulation. Fortunately for the security of simulated beings, a simulation that is too slow with respect to the level that computes it is certainly much less useful and it is likely that most advanced civilizations will instead choose to simulate fewer consciousnesses, but at a faster rate.

Our treatment of the recursive scenario relies on the assumption that, on average, intelligent beings will behave roughly the same no matter on which level of the hierarchy of simulations they exist, including the base reality layer. This is somewhat akin to Bostrom's substrate-independence and bland indifference principles [3], where one assumes the proportion of simulated beings represents our own probability of being simulated, because the type of mind we have (and thus our behaviour) gives us no information as to whether we are real or not. In scenarios where real intelligent beings display the 'pathological' behaviour of killing everyone they encounter while turning all available matter into quantum computers used for simulations, whereas simulated beings do not display that behaviour, the latter will obviously be more numerous. However, given that we take the average over an arbitrarily large amount of space, time, branches of the universe, and even laws of physics compatible with our existence (either as real or simulated beings), these extreme cases do not contribute enough to change our conclusions, unless real and simulated minds are fundamentally different, violating our assumptions. Indeed, the mere *existence* of scenarios where the proportion of simulated beings is high does not mean our final

probability of being simulated is high, just as the existence of scenarios with a low proportion does not mean it is low. Since we have no information indicating which scenario is true (and indeed to a certain extent they might all be true in parallel), only the average matters, which is what our mathematical treatment aims to model.

It is interesting to remark that an important motivation for Bostrom, Musk and other thinkers who began postulating a high probability of our living inside a simulation seems to have been the realization that humanity is approaching the point where it will technologically be able to create its own simulations of intelligent beings (albeit at first probably not quantum ones, nor in the massive amounts that we have been discussing). Once we do have our own simulations, it will instead lend additional credence to our conclusion that we are likely to be real. Indeed, either simulated civilizations cannot (or are very unlikely to) create their own simulations, in which case our having done so will be evidence in favour of our reality, or else the recursive argument applies and equations (2.15) and (2.16) give us a fairly low probability of being simulated, as illustrated by equation (2.17).

## 3. Surveillance from above

If our universe happens to be a deliberate simulation, we might have to worry about being observed by beings who live on the levels closer to reality. As mentioned in §2, we can also worry that they will stop simulating us, intentionally or not, but there is not much we can do about that, except maybe let them observe us and strive to remain entertaining.

If the simulation is flawless and runs on classical computing power, it is fairly obvious that no defence is possible against external observation. Indeed, if that is the case quantum phenomena in our universe are also simulated by classical operations. All information can therefore be copied at will by the simulators without affecting the simulation, and thus without possibility of detection. In fact, this is exactly what we do when we play a video game or perform most of our physics simulations. In addition to not being able to rely on quantum effects, we would also not even be protected by the constraint of no-signalling,[5] because the distance between two points may be completely different in the simulating system and the evolution of the simulation could be paused to allow for interaction between distant components of the system.

For once, it is to our advantage that perfection is so difficult to achieve. If the simulation is classical but imperfect, we might be able to figure it out. As is often seen in video games, computing errors can have a pretty flagrant impact on a simulated world. To illustrate this impact, we can look at collision detection errors, still very frequent despite the growing realism of modern games. When two objects that should collide pass through one another, the resulting 'clipping' can break the player's immersion. The inverse error, where an object's path is blocked even in the absence of an obstacle can be even more problematic, potentially stopping the game's progression. From inside a simulation, such errors leading to incoherent laws of physics might be detected, hinting at the world's artificiality.

Even if the laws of physics are perfectly programmed in the simulation, any real system is imperfect and can be affected by its environment. A good example is radiation flipping bits in a computer's physical memory. This type of error could affect certain simulated physical processes if the deployed error correction schemes are insufficient. Furthermore, if these errors are sufficiently frequent and affect the simulating system in a non-uniform manner (for instance part of the system could be more exposed to the radiation source), it may even be possible to obtain information on the simulating level from inside the simulation.

If quantum phenomena are as difficult to simulate on classical systems as we believe them to be, a simulation containing our world would most probably run on quantum computing power. Unless the simulators are infinitely patient, the slowdown required to simulate quantum effects on classical systems would be prohibitive for all but the simplest simulations. Can we

---

[5]The *no-signalling principle* states that no action taken somewhere can have an *observable* effect anywhere else at a speed faster than that of light.

then rely on restrictions imposed by quantum mechanics to limit the possible surveillance from above? In a way, we can. To start with, the simulators cannot measure too large a quantity of quantum information without destroying the simulation. The structure of materials depends on quantum superposition in the electrons' state space. A measure (even coming from outside the universe) capable of precisely determining their position would (according to the uncertainty principle $\sigma_{x_j}\sigma_{p_j} \geq \hbar/2$) give them enough momentum to destroy any object they constitute. It is thus impossible for the simulators to be completely omniscient about their simulation. Instead of measuring all the quantum information, they could select a small proportion of the qubits to measure. If this sampling is done in a random manner, this could potentially be detected from within the simulation. Indeed, randomly measuring part of the quantum information would induce errors in the simulation's quantum correlations, which we could eventually detect with sufficiently precise experiments.

If Alice and Bob are (unbeknownst to them) simulated beings testing a quantum key establishment protocol ([14], etc.) and they discover noise on their quantum channel, they should (as cryptographers) suppose it is caused by eavesdropping on the channel. If the noise cannot be explained by the equipment used and they discover no spies with access to their channel, they should (as scientists) start asking questions. If Alice, Bob and their colleagues discover that this noise is omnipresent when the precision of the equipment is sufficient to reveal it, despite all attempts to isolate the experiment from the environment, they should conclude that either they have misunderstood the laws of physics in their universe (as in the historical case of Penzias and Wilson detecting the cosmic microwave background as noise on their antenna [15]), or that they are being targeted by surveillance from outside their universe.

The simulators' abilities could however extend much beyond random and generalized surveillance. They could concentrate their surveillance on macroscopic and therefore 'classical' information that exists within the quantum simulation. If human consciousness were an intrinsically quantum phenomenon (while this view is generally associated with pseudoscience, some serious researchers, among them Sir Roger Penrose [16], have proposed models for it, and it remains fairly popular in the more general public), certain details of our thoughts could remain unencoded in macroscopic degrees of freedom and potentially protected against perturbation-free eavesdropping. If instead our minds are fully classical, they are completely vulnerable to such eavesdropping. We believe that only the view where the human mind is fully classical is supported by modern science, and present several arguments to this effect.

To start with, while biological systems are certainly quantum at a microscopic scale and quantum effects contribute to some aspects of brain chemistry, the time scales involved in human thought are more than large enough to make coherence nearly impossible to maintain (especially at the temperature of the human body). Indeed, Tegmark [17] computes a coherence time varying between $10^{-20}$ and $10^{-13}$ s, while the fastest dynamical effects within our neurons take more than $10^{-7}$ s. The interactions between neurons are slower than this by several orders of magnitude (generally below $10^{-3}$ s), so it is extremely unlikely that quantum computation can function based on those, even within a restricted region of the brain.

Beyond the purely physical arguments against quantum computation happening on a large scale in our brains, there are also problems pertaining to its biological evolution. Notably, it is unlikely that quantum computational capabilities would contribute to an organism's evolutionary fitness. While the human eye might well be able to detect single photons [18], nearly all the information we can access through our senses is classical, and all the effects we can have on our environment are also classical. Of course, this is no longer strictly true now that we can perform quantum mechanics experiments, but this is hardly relevant to our evolutionary history. So, as the brain's inputs and outputs are classical, the only fitness advantage that could arise from quantum computing power is an algorithmic speed-up in the treatment of classical information. However, it is fairly obvious that the brain is extremely inefficient at solving problems that have been shown to benefit from a quantum speed-up (factorization, discrete logarithm extraction, black box function inversion, etc.). If the ability to quickly solve such problems mattered in terms of evolutionary fitness, our brains would be much better optimized for them, with or without

quantum capabilities. It is true that many more tasks that benefit from quantum computation have yet to be discovered, but the same reasoning is likely to apply to those as well.

Even if we assume that the addition of quantum computing capabilities to our brain would enhance our evolutionary fitness, another significant problem remains: in order for a trait to arise from natural selection, it cannot be arbitrarily far from pre-existing configurations in genetic phase space. It is for that reason that certain useful structures easily devised by intelligent beings (macroscopic wheel and axle systems for locomotion [19], for example) cannot be found in the animal kingdom. A complex structure can develop gradually if the steps leading to it already confer a fitness advantage. In this way, membranes that enable gliding can eventually evolve into more complex wings, or regions of photoreceptor cells can lead to the formation of eyes. However, it is extremely likely that a cerebral structure enabling quantum computation would be very far (from the state occupied by our distant ancestors) in the genetic phase space, potentially requiring completely different materials from those usually synthesized by living cells. So, unless we can show that intermediate configurations leading to such a structure could by themselves confer a fitness advantage, we can conclude it is nearly impossible for it to have arisen from natural selection.

For the reasons we have presented, the information necessary to describe a human being's thoughts must be essentially classical and very redundant. The external actions we take generally produce consequences on an even larger scale. Our potential simulators could therefore, if they are sufficiently astute, measure only an infinitesimal portion of the simulation's total information while still allowing the reconstruction of our thoughts and actions. In this way, they could identify our precise quantum experiments and avoid creating detectable perturbations on the involved systems. The classical, macroscopic scale on which we live and think thus makes us fundamentally vulnerable. This is no surprise, as almost all practical eavesdropping attacks use in some form or other the redundancy of classical information to copy it without perturbing it in a detectable way (notable exceptions include attacks that exploit weaknesses in poorly implemented quantum cryptographic schemes [20], etc.).

Since our potential simulators could measure (the classical information in) our brain-states, it would be quite difficult to hide information from them. We could envision using the outcome of measurements of quantum systems in order to surprise them (knowing we would ourselves not be able to anticipate the result). We could try to create our own simulated worlds, different in that they would run on a form of encrypted quantum computation to make sure they cannot be spied upon by levels closer to base reality (including our own). If, as discussed in §2, we develop the technological means to project ourselves into the worlds we simulate, we could thus hope to escape the control of our own simulators (apart from their continued ability to end our existence at will by resetting their computing system).

Unfortunately, there is a very general attack that can counter attempts at putting information out of the simulators' reach. As they have physical possession of the systems where all the information is actually encoded and processed, the simulators can (after pausing the evolution of the simulation, if necessary) take part of the information and process it differently from the rest of the simulation.[6] If we model their quantum computer with box and wire diagrams as in figure 2, this corresponds to taking the appropriate wires and connecting them to new boxes implementing a circuit of their choice. At the same time, the simulators can feed different information into the simulation's original boxes (in a way that can depend on the result of the application of the new circuit to the original information). This attack allows the simulators to completely bypass all quantum cryptographic systems we could wish to use, simply by feeding the input into a circuit that does not implement the cryptographic solution. The ability to dynamically change the wiring of the simulation therefore constitutes a potentially insurmountable advantage.

---

[6]Note that in recursive scenarios, higher level simulators would also be able to acquire extensive information about various levels embedded in theirs. They could thus mitigate part of the recursive efficiency loss by intercepting and running themselves the computations (including simulations of further levels) taking place on lower levels, leading to a higher effective average $f_{Eff}$. As it would still be bounded below 1, this higher average efficiency does not compromise our treatment of the recursion.

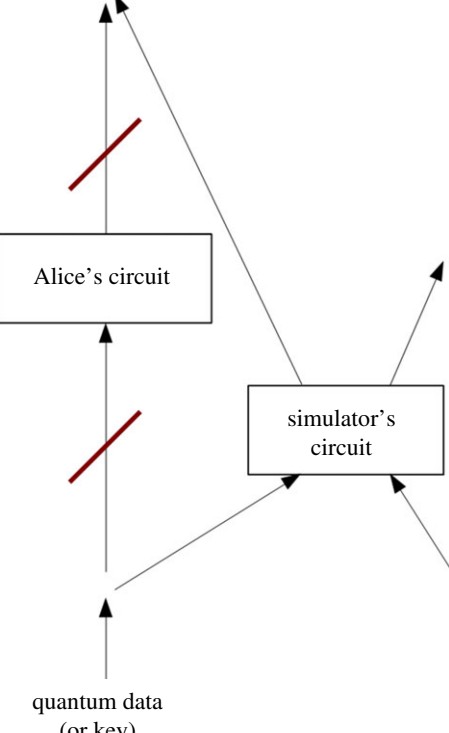

**Figure 2.** A simulated scientist, Alice, attempts to use an encrypted quantum computation system to protect some information (potentially including the thoughts of simulated beings) from external surveillance. Unfortunately, the simulator can thwart that attempt by redirecting the quantum information into a circuit of his choice. Indeed, he can temporarily interrupt the evolution of the simulation, take the appropriate wires in his quantum computer, and connect them to a different circuit (that does not implement Alice's cryptographic protocols). He can use this strategy to substitute not only his own circuits for Alice's, but also his keys and data for Alice's. (Online version in colour.)

In the scenario where we attempt to transfer ourselves into our own simulated worlds, our simulators could intercept the information on which our consciousnesses are encoded and modify or project it into a different environment. We could thus find ourselves in a new simulation under their control, while thinking ourselves securely in the encrypted system we have developed. If the simulators are particularly angry at our attempted escape, they could also send us to a simulated hell, in which case we would at least have the confirmation we were truly living inside a simulation and our paranoia was not unjustified...

## 4. Conclusion

To investigate the possibility that our universe is a deliberate simulation, we use the following assumptions:

(i) Our physics can be efficiently simulated on discrete quantum computers, but not on classical computers.
(ii) Simulated civilizations can create their own simulations once they develop the necessary technology.
(iii) On average, the behaviour of intelligent beings is independent of whether they are real or simulated (recursively or not).

 (iv) In the absence of additional evidence, the total proportion of simulated beings is a good estimate of the probability we live inside a simulation. Note that we inherit this assumption from Bostrom's argument [3]. Even though it is debatable, it becomes nearly impossible to treat this topic in a quantitative manner without it.

Although we argue that Fermi's paradox could potentially be explained by our universe being a simulation of variable complexity, we use a Drake-style equation to show that the proportion of simulated beings, and thus our probability of living inside a simulation, should on average remain fairly low (very likely much under 50%). The main factors in favour of a low probability are the high cost of convincingly simulating a civilization's environment, the unavoidable imperfect efficiency of any computation, and the fact that simulations can be recursive.

If despite this we are simulated after all, we argue that our simulators could eavesdrop on us without significantly perturbing the simulation, even if it is genuinely quantum. While poorly designed attacks could be detected from within the simulation, the (postulated) fundamentally classical nature of our minds and the simulators' ability to adaptively rewire their computing systems would enable them to mount an unbeatable general attack.

**Data accessibility.** This article has no additional data.

**Authors' contributions.** A.B.D. came up with the idea for this paper as part of his PhD thesis under the supervision of G.B. Both authors worked out the basic premises and assumptions during a series of discussions, in particular when A.B.D. visited G.B. at the Institute for Theoretical Studies in Zürich. A.B.D. completed the mathematical analyses and drafted the paper. Subsequently, both authors refined the writing, explanations and conclusions. Both authors gave final approval for publication and agree to be held accountable for the work performed therein.

**Competing interests.** We declare we have no competing interests.

**Funding.** The work of G.B. is supported in part by the Canadian Institute for Advanced Research, the Canada Research Chair program, Canada's Natural Sciences and Engineering Research Council and Québec's Institut transdisciplinaire d'information quantique. Part of this work was supported by the Institute for Theoretical Studies (ITS) at the Eidgenössische Technische Hochschule (ETH), Zürich, when G.B. was one of their Senior Fellows.

**Acknowledgements.** A.B.D. acknowledges Alain Tapp and Louis Salvail for fruitful discussions on topics pertaining to this work. G.B. acknowledges discussions with Stéphane Durand.

# Appendix A. Table of symbols

| | |
|---|---|
| OPS | operations per second |
| $D_{\mathrm{Mat}}$ | maximum computing power density of matter, roughly $10^{50}$ OPS/kg |
| $M_{\mathrm{Bra}}$ | average mass of a human brain, roughly 1.4 kg |
| $P_{\mathrm{Bra}}$ | typical computing power of a human brain, roughly $10^{16}$ OPS |
| Civ | technological level necessary to exploit a 'sizable' fraction of $D_{\mathrm{Mat}}$ |
| $M_{\mathrm{Use}}$ | average equivalent mass in maximally harnessed matter usable by an individual in Civ |
| $R_{\mathrm{Cal}}$ | average number of virtual brains that can be simulated by an individual in Civ |
| $f_{\mathrm{Civ}}$ | fraction of individuals having access to Civ over the history of a civilization |
| $f_{\mathrm{Ded}}$ | fraction of available computing power in Civ dedicated to simulating consciousnesses |
| $N_{\mathrm{Re}}$ | number of real individuals over the history of a civilization |
| $N_{\mathrm{Sim}}$ | number of simulated individuals over the history of a civilization |
| $f_{\mathrm{Re}}$ | fraction of real individuals over the history of a civilization |
| $f_{\mathrm{Sim}}$ | fraction of simulated individuals over the history of a civilization |
| $C_{\mathrm{Env}}$ | proportional computational cost of simulating an individual's environment |
| $f_{\mathrm{Eff}}$ | computational efficiency of a simulation |
| $N_i$ | population native to level $i$ in a recursive chain of simulations |
| $f_{\mathrm{Pop}}$ | average population ratio $N_{i+1}/N_i$ between consecutive simulation levels |

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
