## [Peer Review File · Proceedings. Mathematical, Physical, and Engineering Sciences]

Review History

RSPA-2020-0658.R0 (Original submission)

Review form: Referee 1

Is the manuscript an original and important contribution to its field?

Marginal

Is the paper of sufficient general interest?

Excellent

Is the overall quality of the paper suitable?

Acceptable

Can the paper be shortened without overall detriment to the main message?

Yes

Do you think some of the material would be more appropriate as an electronic appendix?

No

Do you have any ethical concerns with this paper?

No

Recommendation?

Major revision is needed (please make suggestions in comments)

Comments to the Author(s)

This paper explores a question normally associated with movies and sophomore bull sessions: namely, is it probable that the whole universe is a computer simulation? It specifically discusses how computational resource constraints and quantum mechanics interact with that question. It ends with a discussion of how much we could keep secret from the beings simulating us (the answer, unsurprisingly, is "not much").

I found this paper well-written, I had fun reading it, and I think others would have fun reading it too. And I respect the authors' audacity to try to study these matters.

I don't think it's obvious what to do with the paper, but I'm confident of the following: if the paper is rejected from PRSA, it should *not* be because the topic is "too crazy or science-fictiony," but rather, simply because the paper doesn't sufficiently advance the discussion of the topic compared to where it was before.

The heart of the paper is a Drake-equation-style argument, purporting to show that the probability that we live in a computer simulation is not quite as great as Nick Bostrom, Elon Musk, and others have asserted. The idea, basically, is that every simulated universe needs computational resources, but a simulated world (almost by definition) has fewer computational resources available than the world simulating it. So, while it's possible that reality consists of a gigantic tree of simulations inside simulations inside, etc., that tree would quickly "bottom out" as the computing cycles run out.

There were two central points that confused me about this argument.

First, the authors repeatedly assert that the tree of simulations must have finite depth, I don't see how we could possibly know any such thing, if the "base-level reality" were infinite (as for all we know it is). Crucially, even if we agree that every chain of a simulation inside a simulation inside a simulation ... will terminate after a finite number of steps, when the computing resources run out, it's still possible that the tree could contain arbitrarily long chains! This would happen if, for example, the people in the infinite base-level reality were able to run larger and larger finite simulations as time went on. How does this obvious possibility affect the conclusions? I searched in vain for a discussion.

Second, while I didn't follow all the calculations, the key seems to be a decreasing geometric series for the probability of living in an n th-level simulation -- decreasing because the computational resources run out. But isn't there a non-monotone effect? E.g., consider a universe with only a few people, who nevertheless create simulations with many people, who create simulations with even more people. Sure, *eventually* the number of beings at the n th level will start to decrease with n , but at first it goes up! But where is that in the equations? Meanwhile, the whole concluding section, about "keeping secrets from the simulators" using quantum key distribution and the like, feels to me like it could be shortened enormously. Like, isn't it obvious that the simulators are "gods" from our standpoint -- that they can look anywhere they like, for example at the macroscopic observables that describe our own brains and labs, and therefore never need to resort to measuring the quantum states we communicate (even supposing the "gods" are using a quantum computer to simulate us)? Indeed, after a few pages, the paper seems to reach the same totally obvious conclusion ... but why not skip the few pages, and just go straight to the obvious conclusion? :-)

Likewise, the detailed discussion about whether the old question of whether the human brain could be doing quantum computation seems beside the point for this paper -- since even if the brain *were* doing QC, that STILL wouldn't ensure the "privacy" of our thoughts from the simulators! We'd additionally need that our quantumly computed thoughts didn't get recorded in any "classical" degrees of freedom, like the strengths of synapses -- something that just seems

clearly, massively empirically falsified.

DETAILED COMMENTS:

- I think it would help the reader enormously to illustrate the calculations on page 8 with some sample numbers.

- For what it's worth, I think the mention of fractional quantum Hall, and the difficulties of simulating it with quantum Monte Carlo, is totally superfluous and should be removed. This isn't fundamentally different from 500 other situations where we have trouble classically simulating quantum systems! Yes, the authors asserted it had something to do with whether the universe is a simulation, but there's no reason for others to validate that delusion.

The authors constantly use "we note" where I think they mean "we denote by"

I STRONGLY recommend giving author names rather than writing things like "Reference [3]"

bought down  brought down

"A simulated universe with a variable level of complexity could provide an explanation to Fermi's paradox."

Wait, so what about deep sea creatures, and all the other life on Earth that isn't needed for humans, or that wasn't even known to humans before recently? Should we imagine that none of that stuff was simulated in any detail until humans deigned to look at it?

Review form: Referee 2

Is the manuscript an original and important contribution to its field?

Excellent

Is the paper of sufficient general interest?

Excellent

Is the overall quality of the paper suitable?

Good

Can the paper be shortened without overall detriment to the main message?

Yes

Do you think some of the material would be more appropriate as an electronic appendix?

No

Do you have any ethical concerns with this paper?

No

Recommendation?

Accept with minor revision (please list in comments)

Comments to the Author(s)

This is a very interesting and highly original contribution to a line of inquiry that has so far received little scientific inquiry. The question is what are the odds that the phenomena we observe are the result of a computer simulation, and, of this is the case, what would be the

consequences.

The starting point is an argument based on the computational power that is (or seems to be) available in nature. Based on this, it is estimated that a civilization capable of harnessing such power would be in condition to simulate a large number of individuals with weaker computational power, such as ourselves. The authors refine this argument, pointing out that some important variables, such as the complexity of simulating the environment, and the possibility of recursive simulations, were not taken into account.

After these considerations, the authors discuss the scenario in which our observed reality is in fact a simulation. They consider the possibility of using quantum cryptography to hide part of the simulation from external agents, but conclude that the classical nature of our brains makes such hiding impossible. The argument is that, since the external agents can read out our (classical) thoughts without disturbing the simulation, they can find out our plan to encode our information into a quantum-protected computation, and therefore they can rewire such information to a classical simulation of the quantum computation we were planning to perform.

Overall, I found this manuscript a very stimulating reading. Given its uniqueness in a mostly unexplored area, I believe that it will also stimulate the interest of other colleagues. For these reasons, I recommend publication after a few minor points are addressed.

The mentioned points are as follows:

(1) The writing is very clear, and the authors' arguments can be easily followed at each step. However, the global message can sometime be less easy to extract, because the take-home messages of the paper are a bit hidden in the flow of the arguments.

The points where I had this difficulty are as follows:

-the abstract says: "Evaluating loose bounds for certain terms in the equation shows that the probability is unlikely to be as high as previously reported in the literature, especially in a scenario where the simulations are recursive."

In fact, the manuscript presents arguments both for a lower probability (cost of simulating the environment, cost of recursive simulations), and for a higher probability (saving computational resources by simulating only one intelligent civilization). It would be nice would be to add a "Conclusion" section in the end of the paper, with a quick summary of the arguments for a lower probability and those for a higher probability. A table, or a list of bullet points could work well. It may be nice also to anticipate these conclusions in the end of the first paragraph of the Introduction, so that the reader has an idea of what to expect in Section 2.

-the introduction says "As with many things in theoretical computer science, the idea that our world may be a simulation needs revisiting in light of the development of quantum computing." While this is true at the general level, I could not find a specific point in the paper where the specific features of quantum computing factor into the equation. Perhaps the authors referred to this in the last part of the paper, where they discuss the possibility of using a quantum computation to encrypt some of the information in the simulation. An easy way to clarify this is to anticipate the main conclusions in the end of the second paragraph of the introduction.

(2) after Eq. (2.5), perhaps "our mathematical formulation" could be replaced with "the mathematical formulation", because it is later said that the authors challenge this formulation.

Decision letter (RSPA-2020-0658.R0)

15-Dec-2020

Dear Professor Brassard

The Editor of Proceedings A has now received comments from referees on the above paper and would like you to revise it in accordance with their suggestions which can be found below (not including confidential reports to the Editor).

Please submit a copy of your revised paper within four weeks - if we do not hear from you within this time then it will be assumed that the paper has been withdrawn. In exceptional circumstances, extensions may be possible if agreed with the Editorial Office in advance.

Please note that it is the editorial policy of Proceedings A to offer authors one round of revision in which to address changes requested by referees. If the revisions are not considered satisfactory by the Editor, then the paper will be rejected, and not considered further for publication by the journal. In the event that the author chooses not to address a referee's comments, and no scientific justification is included in their cover letter for this omission, it is at the discretion of the Editor whether to continue considering the manuscript.

To revise your manuscript, log into <http://mc.manuscriptcentral.com/prsa> and enter your Author Centre, where you will find your manuscript title listed under "Manuscripts with Decisions." Under "Actions," click on "Create a Revision." Your manuscript number has been appended to denote a revision.

You will be unable to make your revisions on the originally submitted version of the manuscript. Instead, revise your manuscript and upload a new version through your Author Centre.

When submitting your revised manuscript, you will be able to respond to the comments made by the referee(s) and upload a file "Response to Referees" in Step 1: "View and Respond to Decision Letter". Please use this to document how you have responded to the comments, and the adjustments you have made. In order to expedite the processing of the revised manuscript, please be as specific as possible in your response to the referee(s).

IMPORTANT: Your original files are available to you when you upload your revised manuscript. Please delete any unnecessary previous files before uploading your revised version.

When revising your paper please ensure that it remains under 28 pages long. In addition, any pages over 20 will be subject to a charge (£150 + VAT (where applicable) per page). Your paper has been ESTIMATED to be 13 pages.

Once again, thank you for submitting your manuscript to Proc. R. Soc. A and I look forward to receiving your revision. If you have any questions at all, please do not hesitate to get in touch.

Yours sincerely
Raminder Shergill
proceedingsa@royalsociety.org

Reviewer(s)' Comments to Author:
Referee: 1

Comments to the Author(s)

This paper explores a question normally associated with movies and sophomore bull sessions: namely, is it probable that the whole universe is a computer simulation? It specifically discusses how computational resource constraints and quantum mechanics interact with that question. It

ends with a discussion of how much we could keep secret from the beings simulating us (the answer, unsurprisingly, is "not much").

I found this paper well-written, I had fun reading it, and I think others would have fun reading it too. And I respect the authors' audacity to try to study these matters.

I don't think it's obvious what to do with the paper, but I'm confident of the following: if the paper is rejected from PRSA, it should *not* be because the topic is "too crazy or science-fictiony," but rather, simply because the paper doesn't sufficiently advance the discussion of the topic compared to where it was before.

The heart of the paper is a Drake-equation-style argument, purporting to show that the probability that we live in a computer simulation is not quite as great as Nick Bostrom, Elon Musk, and others have asserted. The idea, basically, is that every simulated universe needs computational resources, but a simulated world (almost by definition) has fewer computational resources available than the world simulating it. So, while it's possible that reality consists of a gigantic tree of simulations inside simulations inside, etc., that tree would quickly "bottom out" as the computing cycles run out.

There were two central points that confused me about this argument.

First, the authors repeatedly assert that the tree of simulations must have finite depth, I don't see how we could possibly know any such thing, if the "base-level reality" were infinite (as for all we know it is). Crucially, even if we agree that every chain of a simulation inside a simulation inside a simulation ... will terminate after a finite number of steps, when the computing resources run out, it's still possible that the tree could contain arbitrarily long chains! This would happen if, for example, the people in the infinite base-level reality were able to run larger and larger finite simulations as time went on. How does this obvious possibility affect the conclusions? I searched in vain for a discussion.

Second, while I didn't follow all the calculations, the key seems to be a decreasing geometric series for the probability of living in an n th-level simulation -- decreasing because the computational resources run out. But isn't there a non-monotone effect? E.g., consider a universe with only a few people, who nevertheless create simulations with many people, who create simulations with even more people. Sure, *eventually* the number of beings at the n th level will start to decrease with n , but at first it goes up! But where is that in the equations?

Meanwhile, the whole concluding section, about "keeping secrets from the simulators" using quantum key distribution and the like, feels to me like it could be shortened enormously. Like, isn't it obvious that the simulators are "gods" from our standpoint -- that they can look anywhere they like, for example at the macroscopic observables that describe our own brains and labs, and therefore never need to resort to measuring the quantum states we communicate (even supposing the "gods" are using a quantum computer to simulate us)? Indeed, after a few pages, the paper seems to reach the same totally obvious conclusion ... but why not skip the few pages, and just go straight to the obvious conclusion? :-)

Likewise, the detailed discussion about whether the old question of whether the human brain could be doing quantum computation seems beside the point for this paper -- since even if the brain *were* doing QC, that STILL wouldn't ensure the "privacy" of our thoughts from the simulators! We'd additionally need that our quantumly computed thoughts didn't get recorded in any "classical" degrees of freedom, like the strengths of synapses -- something that just seems clearly, massively empirically falsified.

DETAILED COMMENTS:

- I think it would help the reader enormously to illustrate the calculations on page 8 with some sample numbers.

- For what it's worth, I think the mention of fractional quantum Hall, and the difficulties of simulating it with quantum Monte Carlo, is totally superfluous and should be removed. This isn't fundamentally different from 500 other situations where we have trouble classically simulating quantum systems! Yes, the authors asserted it had something to do with whether the universe is a simulation, but there's no reason for others to validate that delusion.

The authors constantly use "we note" where I think they mean "we denote by"

I STRONGLY recommend giving author names rather than writing things like "Reference [3]"

bought down  brought down

"A simulated universe with a variable level of complexity could provide an explanation to Fermi's paradox."

Wait, so what about deep sea creatures, and all the other life on Earth that isn't needed for humans, or that wasn't even known to humans before recently? Should we imagine that none of that stuff was simulated in any detail until humans deigned to look at it?

Referee: 2

Comments to the Author(s)

This is a very interesting and highly original contribution to a line of inquiry that has so far received little scientific inquiry. The question is what are the odds that the phenomena we observe are the result of a computer simulation, and, of this is the case, what would be the consequences.

The starting point is an argument based on the computational power that is (or seems to be) available in nature. Based on this, it is estimated that a civilization capable of harnessing such power would be in condition to simulate a large number of individuals with weaker computational power, such as ourselves. The authors refine this argument, pointing out that some important variables, such as the complexity of simulating the environment, and the possibility of recursive simulations, were not taken into account.

After these considerations, the authors discuss the scenario in which our observed reality is in fact a simulation. They consider the possibility of using quantum cryptography to hide part of the simulation from external agents, but conclude that the classical nature of our brains makes such hiding impossible. The argument is that, since the external agents can read out our (classical) thoughts without disturbing the simulation, they can find out our plan to encode our information into a quantum-protected computation, and therefore they can rewire such information to a classical simulation of the quantum computation we were planning to perform.

Overall, I found this manuscript a very stimulating reading. Given its uniqueness in a mostly unexplored area, I believe that it will also stimulate the interest of other colleagues. For these reasons, I recommend publication after a few minor points are addressed.

The mentioned points are as follows:

(1) The writing is very clear, and the authors' arguments can be easily followed at each step. However, the global message can sometime be less easy to extract, because the take-home messages of the paper are a bit hidden in the flow of the arguments.

The points where I had this difficulty are as follows:

-the abstract says: "Evaluating loose bounds for certain terms in the equation shows that the probability is unlikely to be as high as previously reported in the literature, especially in a scenario where the simulations are recursive."

In fact, the manuscript presents arguments both for a lower probability (cost of simulating the environment, cost of recursive simulations), and for a higher probability (saving computational resources by simulating only one intelligent civilization). It would be nice would be to add a "Conclusion" section in the end of the paper, with a quick summary of the arguments for a lower probability and those for a higher probability. A table, or a list of bullet points could work well. It may be nice also to anticipate these conclusions in the end of the first paragraph of the Introduction, so that the reader has an idea of what to expect in Section 2.

-the introduction says "As with many things in theoretical computer science, the idea that our world may be a simulation needs revisiting in light of the development of quantum computing." While this is true at the general level, I could not find a specific point in the paper where the specific features of quantum computing factor into the equation. Perhaps the authors referred to this in the last part of the paper, where they discuss the possibility of using a quantum computation to encrypt some of the information in the simulation. An easy way to clarify this is to anticipate the main conclusions in the end of the second paragraph of the introduction.

(2) after Eq. (2.5), perhaps "our mathematical formulation" could be replaced with "the mathematical formulation", because it is later said that the authors challenge this formulation.

Author's Response to Decision Letter for (RSPA-2020-0658.R0)

See Appendix A.

RSPA-2020-0658.R1 (Revision)

Review form: Referee 1

Is the manuscript an original and important contribution to its field?

Acceptable

Is the paper of sufficient general interest?

Excellent

Is the overall quality of the paper suitable?

Good

Can the paper be shortened without overall detriment to the main message?

Yes

Do you think some of the material would be more appropriate as an electronic appendix?

No

Do you have any ethical concerns with this paper?

No

Recommendation?

Accept as is

Comments to the Author(s)

I thank the authors for their responses, and think the paper is now suitable for PRSA.

Review form: Referee 2**Is the manuscript an original and important contribution to its field?**

Excellent

Is the paper of sufficient general interest?

Excellent

Is the overall quality of the paper suitable?

Excellent

Can the paper be shortened without overall detriment to the main message?

Yes

Do you think some of the material would be more appropriate as an electronic appendix?

Yes

Do you have any ethical concerns with this paper?

No

Recommendation?

Accept as is

Comments to the Author(s)

In the revised manuscript, the authors have addressed all my comments and suggestions. The manuscript is now easily accessible and insightful. I am happy to recommend it for publication in PRSA without any further changes.

Decision letter (RSPA-2020-0658.R1)

02-Feb-2021

Dear Professor Brassard

I am pleased to inform you that your manuscript entitled "Probability and consequences of living inside a computer simulation" has been accepted in its final form for publication in Proceedings A.

Our Production Office will be in contact with you in due course. You can expect to receive a proof of your article soon. Please contact the office to let us know if you are likely to be away from e-mail in the near future. If you do not notify us and comments are not received within 5 days of sending the proof, we may publish the paper as it stands.

Open access

You are invited to opt for open access, our author pays publishing model. Payment of open access fees will enable your article to be made freely available via the Royal Society website as soon as it is ready for publication. For more information about open access please visit

<https://royalsociety.org/journals/authors/which-journal/open-access/>. The open access fee for this journal is £1700/\$2380/€2040 per article. VAT will be charged where applicable.

Note that if you have opted for open access then payment will be required before the article is published – payment instructions will follow shortly.

If you wish to opt for open access then please inform the editorial office (proceedingsa@royalsociety.org) as soon as possible.

Your article has been estimated as being 14 pages long. Our Production Office will inform you of the exact length at the proof stage.

Proceedings A levies charges for articles which exceed 20 printed pages. (based upon approximately 540 words or 2 figures per page). Articles exceeding this limit will incur page charges of £150 per page or part page, plus VAT (where applicable).

Under the terms of our licence to publish you may post the author generated postprint (ie. your accepted version not the final typeset version) of your manuscript at any time and this can be made freely available. Postprints can be deposited on a personal or institutional website, or a recognised server/repository. Please note however, that the reporting of postprints is subject to a media embargo, and that the status the manuscript should be made clear. Upon publication of the definitive version on the publisher's site, full details and a link should be added.

You can cite the article in advance of publication using its DOI. The DOI will take the form: 10.1098/rspa.XXXX.YYYY, where XXXX and YYYY are the last 8 digits of your manuscript number (eg. if your manuscript number is RSPA-2017-1234 the DOI would be 10.1098/rspa.2017.1234).

For tips on promoting your accepted paper see our blog post: <https://royalsociety.org/blog/2020/07/promoting-your-latest-paper-and-tracking-your-results/>

On behalf of the Editor of Proceedings A, we look forward to your continued contributions to the Journal.

Sincerely,
Raminder Shergill
proceedingsa@royalsociety.org

Reviewer(s)' Comments to Author:

Referee: 1

Comments to the Author(s)

I thank the authors for their responses, and think the paper is now suitable for PRSA.

Referee: 2

Comments to the Author(s)

In the revised manuscript, the authors have addressed all my comments and suggestions. The manuscript is now easily accessible and insightful. I am happy to recommend it for publication in PRSA without any further changes.

Appendix A

Dear Raminder

Thank you for the referee reports on our paper
“Probability and consequences of living inside a computer simulation”
submitted for possible publication in the *Proceedings of the Royal Society A*.
We found that most of the comments were useful in improving our work.

We have thoroughly revised our manuscript to take account of most of the referees’ suggestions. Please find below our point-by-point answers to the referees, with an explanation when we felt that following the suggestions was not appropriate. In what follows, the referees’ text is set in blue and our responses in black.

Referee 1

Comments to the Author(s)

This paper explores a question normally associated with movies and sophomore bull sessions: namely, is it probable that the whole universe is a computer simulation? It specifically discusses how computational resource constraints and quantum mechanics interact with that question. It ends with a discussion of how much we could keep secret from the beings simulating us (the answer, unsurprisingly, is "not much").

Right.

I found this paper well-written, I had fun reading it, and I think others would have fun reading it too. And I respect the authors' audacity to try to study these matters.

Thank you!

I don't think it's obvious what to do with the paper, but I'm confident of the following: if the paper is rejected from PRSA, it should *not* be because the topic is "too crazy or science-fictiony,"

Thank you for saying this. Obviously, we agree.

but rather, simply because the paper doesn't sufficiently advance the discussion of the topic compared to where it was before.

While it is hard to evaluate how much remains to be understood and how difficult further progress will be in this mostly unexplored topic, we think that our paper presents significant advances. For instance, we study recursive scenarios, in which simulated civilizations can create their own simulations when they have reached sufficient technological sophistication, and we find that this decreases rather than increases the proportion of simulated beings. Also, we give a new potential, if unlikely, resolution of Fermi's paradox. Furthermore, we offer a more in-depth and interconnected treatment of previously discussed ideas that should help other researchers make novel contributions.

We point out that the other referee concurs with us, saying "This is a very interesting and highly original contribution to a line of inquiry that has so far received little scientific inquiry" and "Overall, I found this manuscript a very stimulating reading. Given its uniqueness in a mostly unexplored area, I believe that it will also stimulate the interest of other colleagues".

The heart of the paper is a Drake-equation-style argument, purporting to show that the probability that we live in a computer simulation is not quite as great as Nick Bostrom, Elon Musk, and others have asserted.

Right.

The idea, basically, is that every simulated universe needs computational resources, but a simulated world (almost by definition) has fewer computational resources available than the world simulating it. So, while it's possible that reality consists of a gigantic tree of simulations inside simulations inside, etc., that tree would quickly "bottom out" as the computing cycles run out.

Correct again.

There were two central points that confused me about this argument.

First, the authors repeatedly assert that the tree of simulations must have finite depth, I don't see how we could possibly know any such thing, if the "base-level reality" were infinite (as for all we know it is). Crucially, even if we agree that every chain of a simulation inside a simulation inside a simulation ... will terminate after a finite number of steps, when the computing resources run out, it's still possible that the tree could contain arbitrarily long chains! This would happen if, for example, the people in the infinite base-level reality were able to run larger and larger finite simulations as time went on. How does this obvious possibility affect the conclusions? I searched in vain for a discussion.

I'm afraid you confuse infinite with unbounded. We agree that there is no a priori bound on the simulation depth, and that it could keep increasing as civilizations manage to harness larger and larger regions of space. However, no simulation will ever have infinite depth. As an analogy, consider the statement according to which no natural number is infinite. This does not preclude the existence of arbitrarily large integers.

This being said, we recognize that other readers could get confused as well. For this reason, this issue is now discussed in footnote 4, which explains that the chains cannot be infinite in length. Even though they may grow to an arbitrarily long length, this does not affect the conclusions. Even in the most extreme case in which arbitrarily long chains dominate, it merely causes the bound of the infinite geometric series of Eq. (2.14) to be attained. To take this into account, some strict inequalities in the equations have been modified to "smaller than or equal to" in order to give some extra manoeuvring room in that direction. This causes no changes in the conclusions.

Second, while I didn't follow all the calculations, the key seems to be a decreasing geometric series for the probability of living in an n th-level simulation -- decreasing because the computational resources run out. But isn't there a non-monotone effect? E.g., consider a universe with only a few people, who nevertheless create simulations with many people, who create simulations with even more people. Sure, *eventually* the number of beings at the n th level will start to decrease with n , but at first it goes up! But where is that in the equations?

This is now discussed in two new paragraphs at the end of Section 2. While non-monotone effects are possible, the average number of beings per level decreases in a monotone (and geometric) fashion, once integrated over all appropriate parameters. The assumptions justifying this are now more explicit in section 2 and reiterated in the new Conclusion section.

Meanwhile, the whole concluding section, about "keeping secrets from the simulators" using quantum key distribution and the like, feels to me like it could be shortened enormously. Like, isn't it obvious that the simulators are "gods" from our standpoint -- that they can look anywhere they like, for example at the macroscopic observables that describe our own brains and labs, and therefore never need to resort to measuring the quantum states we communicate (even supposing the "gods" are using a quantum computer to simulate us)? Indeed, after a few pages, the paper seems to reach the same totally obvious conclusion ... but why not skip the few pages, and just go straight to the obvious conclusion? :-)

While our conclusion to this part of the paper is certainly not unexpected, we do not think it is so obvious. Given the questions and comments we have received on the subject during discussions with colleagues, presentations of partial work, and ABD's thesis defence, we are fairly confident that it is at least not obvious to all potential readers. If a simulation is truly quantum, it is not immediately clear that no strategy exists that can leverage the no-cloning theorem to hide at least some information from the simulators, or force them to create a measurable amount of perturbation during their eavesdropping.

Likewise, the detailed discussion about whether the old question of whether the human brain could be doing quantum computation seems beside the point for this paper -- since even if the brain *were* doing QC, that STILL wouldn't ensure the "privacy" of our thoughts from the simulators! We'd additionally need that our quantumly computed thoughts didn't get recorded in any "classical" degrees of freedom, like the strengths of synapses -- something that just seems clearly, massively empirically falsified.

This may be obvious to you and we do not disagree. Nevertheless, we do not feel that a viewpoint championed by luminaries as illustrious as Nobel Prize laureate Sir Roger Penrose FRS can be so casually dismissed. A view in which consciousness itself is fundamentally quantum demands that at least some of its components not always be recorded on the macroscopic level. Should that be true, it could conceivably partially protect simulated consciousnesses against perturbation-free eavesdropping.

Furthermore, the number of questions we have seen with respect to quantum consciousness shows that it is not yet falsified enough to convince all potential readers. While it is empirically obvious that at least a part of our thoughts are encoded in classical degrees of freedom, it is certainly not yet experimentally demonstrated that it is true of everything that contributes to them. Until we have a way to record the entirety of a person's stream of consciousness and a good enough model of their mind to feed it into to predict their future behaviour very accurately, we cannot consider we have irrefutable experimental proof that everything about our thoughts is encoded classically. Nevertheless, we still believe that the theoretical arguments we present in our paper (not all of which are original) make a sufficiently convincing case for a completely classical consciousness.

For all these reasons, we consider that our discussion of these issues in our paper is essential to make a convincing case for our conclusions. Furthermore, we rewrote most of the paragraph that introduces this issue in order to make it more obvious that it is relevant.

DETAILED COMMENTS:

- I think it would help the reader enormously to illustrate the calculations on page 8 with some sample numbers.

Good point. We added Eq (2.17) as one such example.

- For what it's worth, I think the mention of fractional quantum Hall, and the difficulties of simulating it with quantum Monte Carlo, is totally superfluous and should be removed. This isn't fundamentally different from 500 other situations where we have trouble classically simulating quantum systems! Yes, the authors asserted it had something to do with whether the universe is a simulation, but there's no reason for others to validate that delusion.

Agreed. All mentions of the fractional quantum Hall effect have been deleted.

The authors constantly use "we note" where I think they mean "we denote by"

Agreed. We have made this correction throughout.

I STRONGLY recommend giving author names rather than writing things like "Reference [3]"

Agreed again. This has been fixed throughout.

bought down  brought down

Done. Thanks.

"A simulated universe with a variable level of complexity could provide an explanation to Fermi's paradox."

Wait, so what about deep sea creatures, and all the other life on Earth that isn't needed for humans, or that wasn't even known to humans before recently? Should we imagine that none of that stuff was simulated in any detail until humans deigned to look at it?

While this is possible, both the amount of computational resources that can be saved this way (compared to simulating the rest of the observable universe) and the degree to which the simulation (of deep sea ecosystems, etc.) can be simplified without affecting the outcome, are low. This is now mentioned in the paper (top of page 6).

Referee 2:

Comments to the Author(s)

This is a very interesting and highly original contribution to a line of inquiry that has so far received little scientific inquiry

Thank you. We appreciate this appraisal.

The question is what are the odds that the phenomena we observe are the result of a computer simulation, and, of this is the case, what would be the consequences.

Right.

The starting point is an argument based on the computational power that is (or seems to be) available in nature. Based on this, it is estimated that a civilization capable of harnessing such power would be in condition to simulate a large number of individuals with weaker computational power, such as ourselves. The authors refine this argument, pointing out that some important variables, such as the complexity of simulating the environment, and the possibility of recursive simulations, were not taken into account.

Correct again.

After these considerations, the authors discuss the scenario in which our observed reality is in fact a simulation. They consider the possibility of using quantum cryptography to hide part of the simulation from external agents, but conclude that the classical nature of our brains makes such hiding impossible. The argument is that, since the external agents can read out our (classical) thoughts without disturbing the simulation, they can find out our plan to encode our information into a quantum-protected computation, and therefore they can rewire such information to a classical simulation of the quantum computation we were planning to perform.

Again correct.

Overall, I found this manuscript a very stimulating reading. Given its uniqueness in a mostly unexplored area, I believe that it will also stimulate the interest of other

colleagues. For these reasons, I recommend publication after a few minor points are addressed.

Thank you so much for such a warm recommendation in favour of publication!

The mentioned points are as follows:

(1) The writing is very clear, and the authors' arguments can be easily followed at each step. However, the global message can sometime be less easy to extract, because the take-home messages of the paper are a bit hidden in the flow of the arguments.

You are right. Thank you for pointing it out. We have made it clear from the onset with clear statements concerning our "take-home messages" with added sentences at the end of both paragraphs in the Introduction. Furthermore, we added a Conclusion section to make sure the take-home messages are well understood by the readers.

The points where I had this difficulty are as follows:

-the abstract says: "Evaluating loose bounds for certain terms in the equation shows that the probability is unlikely to be as high as previously reported in the literature, especially in a scenario where the simulations are recursive."

Correct. This is indeed our main take-home message.

In fact, the manuscript presents arguments both for a lower probability (cost of simulating the environment, cost of recursive simulations), and for a higher probability (saving computational resources by simulating only one intelligent civilization).

We strive to give a fair hearing to both sides of the issues. We made modifications to ensure that our readers will know where we stand once all arguments have been accounted for.

It would be nice would be to add a "Conclusion" section in the end of the paper, with a quick summary of the arguments for a lower probability and those for a higher probability. A table, or a list of bullet points could work well.

Agreed. Done. With a bullet points list of our assumptions.

It may be nice also to anticipate these conclusions in the end of the first paragraph of the Introduction, so that the reader has an idea of what to expect in Section 2.

Agreed. Done.

-the introduction says "As with many things in theoretical computer science, the idea that our world may be a simulation needs revisiting in light of the development of quantum computing."

Correct. Note that we removed the word “theoretical” from the sentence above.

While this is true at the general level, I could not find a specific point in the paper where the specific features of quantum computing factor into the equation. Perhaps the authors refereed to this in the last part of the paper, where they discuss the possibility of using a quantum computation to encrypt some of the information in the simulation. An easy way to clarify this is to anticipate the main conclusions in the end of the second paragraph of the introduction.

We have followed the referee's suggestion of anticipating the conclusions in the introduction and have made it clearer that, in addition to affecting our discussion of eavesdropping from outside the simulation, the quantum nature of our physics, and therefore of the computing power that is probably required to simulate it, restricts the type of simulation we could be in. If we instead lived in a universe with laws of physics that never require (or enable) much more computing power than our brains do, the recursive simulation scenarios would become much less likely, and the "adversarial" ones optimized to drive up the proportion of simulated beings much more likely.

Specifically, we added the following discussion to our paper: “Thus, while the quantum nature of our universe, and probably of the computing power necessary to simulate it, does not change our equations (which would be as valid for classical worlds simulated on classical computers, provided their classical laws of physics can still support a gigantic R_{Cal}), it restricts the type of simulation in which we could likely be. This comes into play as we consider what happens when simulated worlds behave similarly to those from which they are simulated.”

(2) after Eq. (2.5), perhaps "our mathematical formulation" could be replaced with "the mathematical formulation", because it is later said that the authors challenge this formulation.

Agreed. Done.

And thank you for your insightful comments.

- Alexandre Bibeau-Delisle and Gilles Brassard FRS